# Pneumococcal carriage in adults aged 50 years and older in outpatient health care facility during pandemic COVID-19 in Novi Sad, Serbia

Vladimir Petrović[1,2☯], Mioljub Ristić[1,2☯]*, Biljana Milosavljević[2], Milan Djilas[2], Miloš Marković[3]

1 Faculty of Medicine, Department of Epidemiology, University of Novi Sad, Novi Sad, Serbia, 2 Institute of Public Health of Vojvodina, Novi Sad, Serbia, 3 Faculty of Medicine, Department of Immunology, Institute of Microbiology and Immunology, University of Belgrade, Belgrade, Serbia

☯ These authors contributed equally to this work.
* mioljub.ristic@mf.uns.ac.rs

**Data Availability Statement:** All relevant data are within the paper and its Supporting Information files.

## Abstract

### Background

Data related to carriage of *Streptococcus pneumoniae* (Spn) and antimicrobial resistance patterns in middle-aged and older adults are limited. We assessed the carriage of Spn, and its antibiotic resistance patterns, among participants ≥50 years of age living in the city of Novi Sad during the second year of COVID-19 pandemic.

### Methods

Analysis of prospectively collected data among participants with or without symptoms of upper respiratory tract infection who visited their elected physicians in the Primary Health Care Centre of Novi Sad (outpatient facility) was conducted from May 18, 2021 to December 7, 2021. Both nasopharyngeal (NP) and oropharyngeal (OP) samples from each participant were collected.

### Results

A total of 1042 samples from 521 study subjects (1 NP and 1 OP sample from each person) were collected. Sixteen samples from the same number of persons (3.1%, 95% confidence interval: 1.76%-4.94%) were culture positive for the presence of Spn. Overall, the median age of study participants was 71 years (range, 50–93 years; 90th percentile, 77 years), and most (197/521, 37.8%) of them were 70–79 years of age. A majority of the study subjects were: females (324/521; 62.2%), sampled during May and June 2021 (376/521, 72.2%), those who did not have contact with children aged 0–10 years in the family (403/521; 77.4%), without smokers in the household (443/521; 85.0%), and those who did not receive vaccine against Spn (519/521; 99.6%). Out of 16 Spn positive samples, for six participants, Spn carriage serotypes were obtained and there were four vaccine (6A, 11A, 15B, and 18C)

**Funding:** This work supported through research grants received by Merck Sharp & Dohme doo (grant number 60164). The funders had no role in study design, data collection and analysis, decision to publish, or preparation of the manuscript.

**Competing interests:** VP acts as principal investigator for investigator initiated sponsored studies related to the topic conducted on behalf of the Faculty of Medicine, University of Novi Sad, for which the Faculty obtained research grants from Merck Sharp & Dohme doo. There are no patents, products in development or marketed products associated with this research to declare. This does not alter our adherence to PLOS ONE policies on sharing data and materials.

serotypes, and two (6C and 35F) non-vaccine serotypes. Remaining 10 (62.50%) samples were non-typeable isolates of pneumococci. Among four vaccine serotypes, two (6A and 18C) were represented in PCV13, and 18C along with the other two (11A and 15B) in PPSV23 vaccine. The highest level of resistance of Spn isolates was observed for erythromycin, (10 or 62.50%), and tetracycline, (7 or 43.75%), one isolate showed resistance to penicillin, ampicillin, and amoxicillin/amoxicillin-clavulanic acid, while none of them were resistant to ceftriaxone, trimethoprim/sulfamethoxazole and levofloxacin. There were three multi-drug resistant isolates; one was identified as 6C (non-vaccine serotype), and two other were non-typeable isolates of Spn.

## Conclusions

In this first study conducted in Serbia on Spn carriage in adults ≥50 years of age, we found low prevalence of Spn carriage and identified 6 serotypes of Spn, four of which were represented in vaccines. These results may support future Spn colonization studies among middle-aged and older adults.

## Introduction

Carriage of *Streptococcus pneumoniae* (Spn) is a prerequisite for the disease development and serves as a reservoir for the spread of infection within the community. Pneumoccocal carriage can lead to invasive pneumococcal disease (IPD) such as meningitis, septicaemia and pneumonia, as well as milder but more common illnesses such as sinusitis and otitis media [1]. The human nasopharynx and oropharynx are the primary reservoir of Spn [1–3]. On average, about 75% of cases of IPD, and 83% of cases of pneumococcal meningitis, occur in children under 2 years of age, but the incidence and age distribution of cases may vary by country (sensitivity of case ascertainment and surveillance), study method and socio-economic status within and between countries [1,4]. IPD and community-acquired pneumonia both show a seasonal and climatic trend i.e. they coincide with the seasonal circulation of influenza and respiratory syncytial viruses, suggesting that these viruses predispose to pneumococcal infection [1,5,6].

Data on carriage among adults are scarce, contrary to vast information available on Spn carriage among children [4,7]. Despite the limited data, it is known that the rate of Spn carriage in adults is low, compared to the rate in children [7]. Children may represent a source of pneumococcal transmission (i.e. horizontal spread) to adults in the same household [7,8]. Although the IPD is most commonly recognised among children, due to greater susceptibility to infection as well as to existing comorbidities such as chronic illnesses, the incidence of IPD rises dramatically after the age of 50 years [9]. In addition, in immunocompetent persons with underlying medical conditions the incidence of pneumococcal infections may be as high as 176–483 per 100,000, while the incidence among patients with immunocompromising conditions has been even higher and ranges from 342 to 2,031 per 100,000 persons [10]. Importantly, the crude incidence of IPD is expected to rise with increase in age and is highest in the elderly populations [11].

There are no data on carriage of Spn among adults in Serbia. Surveillance of IPD in our country is based on reporting of clinically diagnosed illness with different clinical presentations (pneumonia, sepsis, bacterial meningitis) with etiological agent being identified in a

small number of cases [12]. Previously published results from the study performed in the city of Novi Sad where IPD surveillance was conducted showed that pneumococcal conjugated vaccines (PCVs), i.e. PCV7, PCV10, and PCV13 covered 48.6%, 54.3%, and 84.8% isolates, respectively, while pneumococcal polysaccharide vaccine (PPSV23) covered 89.5% of the total number of Spn isolates in all age groups. More precisely, Spn serotypes included in PCV13, and PPSV23 covered 87.1%, and 90.3%, of the total number of Spn isolates in adults 50–64 years of age, respectively, and 77.8%, and 85.2%, of the total number of Spn isolates in adults ≥65 years of age, respectively [12].

The good quality epidemiological data of Spn carriage in developing countries are the main precondition for implementing appropriate vaccination strategies and for evaluation of the impact of immunization [1,4]. Introduction of PCVs substantially reduced the prevalence of vaccine serotypes carriage as well as the presence of antibiotic-resistant serotypes among both, vaccinated and unvaccinated people [11,13]. However, surveillance of Spn carriage and IPD in our territory is mainly passive, and there were no investments in the longitudinal surveillance systems required to determine effectiveness of the PCV immunisation programme. According to the Law on Protection of Population against Communicable Diseases and the National Immunization Program in Serbia [14,15], the immunization of adults is mandatory ("free of charge") for high-risk groups (those with underlying medical conditions), and for persons older than 65 years in specific vulnerable settings (nursing home residences and other long-term institutions), while it is recommended for all healthy adults ≥65 years of age. Immunization for aforementioned indications in Serbia are provided with conjugate PCV13 or/and poly-saccharide PPSV23 pneumococcal vaccines [16]. Immunization coverage with pneumococcal vaccines among middle-aged and older adults in our country is very low including among those that are eligible for vaccination based on clinical indications [12], and it is still unknown to which extent immunization may provide benefits to elderly in reducing Spn carriage and disease [9].

The objective of this study was to obtain a baseline measurement of Spn carriage, to assess the antibiotic resistance patterns, and associated risk factors of Spn carriage among subjects ≥50 years of age in the city of Novi Sad (the capital and the administrative centre of Autonomous Province of Vojvodina in Serbia). Additional benefit of the research is that it was conducted during the COVID-19 pandemic in 2021.

## Materials and methods

### Study site, design, and participants

A prospective study on the serotype and antibiotic resistance of Spn was performed in middle-aged and older adults who visited their elected physicians in an outpatient facility, the Primary Health Care Centre (PHCC) of Novi Sad. This was an ongoing, population-based, active surveillance study, based on voluntary participation. Research was conducted in the city of Novi Sad, which is the capital and the largest city in Vojvodina with population of 360.925 inhabitants.

In accordance with the sample size calculation for a study estimating a population prevalence [17], we planned to include between 350 and 500 participants in the study and collect samples from them (one nasopharyngeal-NP and one oropharyngeal-OP sample from each participant). We presumed to detect pneumococci in up to 5% (2% error, 95% CI) of tested subjects. In addition, the total number of adults ≥50 years of age in Novi Sad was estimated to be around 120.000, and therefore we expected to include in the study a number of subjects that corresponds to 0.4% of the total targeted population. Recruitment period had been previously scheduled for the period from June 1, 2020 to May 31, 2021, but due to COVID-19 pandemic waves, it was delayed and started on May 18, 2021 and lasted until December 7, 2021.

The research was carried out by the Medical Faculty of Novi Sad in collaboration with the Institute of Public Health of Vojvodina (IPHV), Centre for Disease Control and Prevention and the Centre for Microbiology, and PHCC of Novi Sad (General medicine Department).

## Inclusion/Exclusion criteria

Participants included in this study were selected and sampled on a daily basis by the physicians at the outpatient settings in PHCC of Novi Sad. Eligible participants were ≥50 years of age, visiting their elected physician at PHCC as a part of their periodic systematic health examination regardless whether they had or did not have any symptoms of upper respiratory tract infection. However, participants were excluded if they lived outside the city of Novi Sad, or if they used antibiotics within two weeks prior to sampling and/or had a confirmed immunodeficiency (haematological malignancies, inherited immunodeficiency, HIV, post-splenectomy status, cancer chemotherapy). Only one participant per household was enrolled. At the admission, physicians interviewed all included participants through face-to-face structured interviews.

First, verbal and written explanation of the aim of the research (**S1 Appendix**) were given to each participant. Then they signed the written informed consent (**S2 Appendix**) in order to be enrolled as participants. After that, study-related procedures were performed. Personal and confidential information obtained from participants were removed, except for their demographic information, including date of sampling, age, gender, number and date of potential application of PCV13 or/and PPSV23 vaccine/s, and two potential risk factors for Spn carriage: contact with children aged 0–10 years residing in the household of participants, and smoking habits of members of the family (**S3 Appendix**).

Characteristics of all study subjects acquired by questionnaire, the results of laboratory testing of samples of each participant and the final characterization of Spn serotypes were entered in a database specially designed for this research. Immediately after the testing was finished in the IPHV, the results of the laboratory tested samples were sent to the elected physicians to inform their patients.

## Sample processing

Considering the recommendation of the World Health Organization (WHO) for adults, both NP and OP samples were collected from each participant enrolled in the study. Carriage determination from the nasopharynx is more sensitive than from the oropharynx for pneumococci, but inclusion of OP may increase the sensitivity of the procedure [18]. The sterile cotton-tipped wire swab was inserted into the anterior nares or oropharynx, gently rubbed their walls and immediately placed in the surrounding skim milk tryptone-glucose-glycerol (STGG) transport medium (Copan Venturi Transystem, Brescia, Italy). The samples, together with the previously completed Survey questionnaires, were transported to the Centre for Microbiology of IPHV within eight hours. All swabs were handled and systematically tested following WHO Pneumococcal Carriage Working Group the standard operating procedures [18]. The primary outcomes were colonization by Spn (culture) and Staphylococcus aureus (culture). Swabs were analysed by 200 μl of swab-inoculated STGG medium and were transferred to 5.0 ml Todd Hewitt broth containing 0.5% yeast extract (THY) and 1 ml of rabbit serum and incubated at 35–37˚C for six hours. Cultured broth was plated on Columbia agar supplemented with 5% sheep blood and gentamycin 5 mg/l and incubated in 5% $CO_2$ at 35–37˚C. After 18–24 hours of incubation, plates were examined for the appearance of alpha-haemolytic colonies resembling streptococci. When alpha-hemolytic colonies growth was noticed on the plate, a single colony was re-cultured, Gram-stained and tested for susceptibility to optochin and bile solubility tests.

## Molecular serotyping

**DNA extraction.** To obtain genomic deoxyribonucleic acid (DNA) extracts for PCR reactions, DNA was isolated from fresh overnight cultures [19,20]. An overnight growth of blood agar plate was suspended in 300 μl of 0.85% NaCl, heated to 70˚C for 15 min, spun for two minutes and supernatant was removed. Pellet was suspended in 50 μl TE buffer with an addition of 10 μl mutanolysin and 8 μl of hyaluronidase, kept at 37˚C for 30 min, heated for 10 minutes at 100˚C and centrifuged for 4 minutes. The resulting supernatant (in a volume of 2,5 μl) was used as DNA template. Extracts were stored at -20˚C until PCR testing.

*Identification of S. pneumonia.* To avoid misidentification of Spn like viridians group streptococci, a Real-Time PCR Assay targeting *lytA* (autolysin) a gene specific for Spn, has been used [21]. Identification of Spn was achieved by PCR through the amplification of the the *lytA* gene using primers described by Nagai K, et al. [22], and Gholamhosseini-Moghaddam T, et al. [23]. Positive samples were further analysed for serotype determination.

*Serotype identification.* Conventional multiplex PCR assays were performed as a series of eight multiplex reactions, using CDC recommended schemes and primers for pneumococcal serotype deduction with the possibility to identify 41 different serotype of Spn [24]. Additional analysis was performed using the series of seven multiplex PCR reactions designed by Jourdain S, et al. [25]. Each reaction included a primer pair for *cpsA* gene as internal control. Further differentiation between 6A, 6B and 6C serotypes was performed using procedures and primers desribed by Jin P and the coworkers. [26]. In order to confirm the results of multiplex PCR we used standard strains: ATCC6305, ATCC49619, ATCC6303 and ATCC700677. The PCR products were analysed on 2% agarose gels, stained with ethidium bromide. Gel images were recorded by BioDocAnalyze system (Analytik Jena, Jena, Germany) [19,27]. If isolates were tested positive with optochin and *lytA* PCR, but further were negative for *cpsA gene* (acting as an IC) and serotype specific sequences in all serotyping PCR reactions, then they were considered as non-typeable isolates.

## Antibiotic resistance testing

Antibiotic resistance of Spn isolates to the set of antibiotics were determined by the Kirby-Bauer disc diffusion technique, according to the European Committee on Antimicrobial Susceptibility Testing (EUCAST) recommendations [28].

## Statistical analysis

Descriptive statistics (mean, median, and standard deviation or percentages where appropriate) were used to summarize the data obtained from interviews. All data, including demographic, health variables, vaccination status as well as antimicrobial resistance patterns of Spn isolates were presented as proportions. The prevalence of Spn carriage as the proportion of participants whose swabs cultures were positive for Spn and corresponding 95% confidence intervals (Cis) were estimated. Categorical variables were compared using chi-square or Fisher's exact test as appropriate. All descriptive analyses were performed using SPSS software tool (version 22) MedCalc for Windows, version 12.3.0 (MedCalcSoftware, Mariakerke, Belgium). Statistical significance was set at the value of p<0.05.

## Ethical consideration

The study protocol was reviewed and approved by the Ethics Committees of the IPHV, Novi Sad the Faculty of Medicine, University of Novi Sad. No authors of this study were involved in

the treatment of the patients included in the analysis, and all data were anonymized before the authors accessed it.

## Results

### Socio-demographic characteristics

A total of 1042 samples from 521 study subjects were collected (1 NP and 1 OP sample from each participant). Sixteen samples from the same number of persons (3.1%, 95% confidence interval: 1.76%-4.94%) were culture positive for the presence of Spn. The participant characteristics are described in Table 1. Median age of study participants was 71 years (range, 50–93 years; 90th percentile, 77 years), and most of the participants included in this research were 70–79 years of age (197/521, 37.8%). A majority of the study subjects were: females (324/521; 62.2%), sampled during May and June 2021 (376/521, 72.2%), those who did not have contact with children aged 0–10 years in the family household (403/521; 77.4%), subjects without smokers in the household (443/521; 85.0%), and those who did not receive vaccine against Spn (519/521; 99.6%). There were also 43 (8.3%) participants with *Staphylococcus aureus* NP carriage, but none of them also harboured Spn.

Among 16 Spn positive samples, 14 were detected from NP, and only two from OP specimens. Out of those 16 Spn isolates, serotypes were determined for six of them: four serotypes present in pneumococcal vaccine (6A, 11A, 15B, and 18C), and the remaining two (6C and 35F) were non-vaccine serotypes. A total of 10 (62.5%) samples were non-typeable isolates of pneumococci. Only two participants who were not colonized with Spn had previously received PCV13 vaccine, while none of the participants in our study had received PPSV23.

We further assessed whether there were any association between the characteristics of participants ≥50 years of age and the carriage of Spn in their NP or OP. Prevalence of Spn positive isolates was significantly higher (p<0.001) during May, June, October and December comparing to a period between July and September. Other observed differences were not statistically significant (p>0.05) (Table 2).

Participants with determined Spn carriage serotypes were between 65 and 81 years of age, with equal gender distribution, and were mainly detected during May 2021. Two of them had a smoker in their household, and half of them did not have contact with children aged 0–10 years in their family. All Spn isolates with determined serotypes were isolated from NP swabs. Among four vaccine serotypes, two (6A and 18C) were represented in PCV13, and 18C along with other two (11A and 15B) in PPSV23 vaccine. In addition, there were 10 non-typeable Spn isolates. Participants with non-typeable Spn carriage isolates were between 50 and 80 years of age, and mainly detected during June 2021. Two of them had a smoker in the household, and also two participants had contact with children aged 0–10 years in their families. With the exception of two participants for whom Spn was isolated from OP specimens, all others were isolated from NP swabs (Table 3). There were no participants vaccinated against pneumococcus with Spn carriage.

### Antimicrobial resistance patterns of *Streptococcus pneumoniae* isolates

All 16 Spn isolates were included in the resistance pattern analysis. The highest level of resistance of Spn isolates was observed for erythromycin, 10 (62.50%), and tetracycline, 7 (43.75%). As for the serotypes, one 6A and one 6C as well as eight non-typeable isolates showed resistance to erythromycin, and by one of 6C and 35F serotypes as well as five non-typeable ones showed resistance to tetracycline. None of the isolates, regardless of the serotype, were resistant to ceftriaxone, trimethoprim/sulfamethoxazole and levofloxacin, while one non-typeable isolate displayed resistance to penicillin, ampicillin, and amoxicillin/amoxicillin-clavulanic acid

**Table 1. Characteristics of the study participants (N = 521).**

| Characteristics | N | % |
|---|---|---|
| **Age (years), mean ± SD [median]** | **70.17±9.08 [71]** | **NA** |
| **Age group** | | |
| 50–59 | 70 | 13.4 |
| 60–69 | 165 | 31.7 |
| 70–79 | 197 | 37.8 |
| ≥80 | 89 | 17.1 |
| **Gender** | | |
| Male | 197 | 37.8 |
| Female | 324 | 62.2 |
| **Month of study entry during 2021** | | |
| May | 190 | 36.5 |
| June | 186 | 35.7 |
| July | 3 | 0.6 |
| August | 12 | 2.3 |
| September | 53 | 10.2 |
| October | 74 | 14.2 |
| December | 3 | 0.6 |
| **Risk factors for pneumococcal carriage** | | |
| Number of children aged 0–10 years in the family | | |
| 0 | 403 | 77.4 |
| 1–2 | 112 | 21.5 |
| ≥3 | 6 | 1.2 |
| Smoker in the household | | |
| Yes | 78 | 15.0 |
| No | 443 | 85.0 |
| **Carriage of *Streptococcus pneumoniae*** | | |
| Negative | 505 | 96.9 |
| Positive in NP | 14 | 2.7 |
| Positive in OP | 2 | 0.4 |
| **Colonization with *Streptococcus pneumoniae*** | 16 | 3.1 |
| *Vaccine serotypes* | | |
| 6A | 1 | 6.3 |
| 11A | 1 | 6.3 |
| 15B | 1 | 6.3 |
| 18 C | 1 | 6.3 |
| Subtotal | 4 | 25.0 |
| *Non-vaccine serotypes* | | |
| 6C | 1 | 6.3 |
| 35F | 1 | 6.3 |
| Subtotal | 2 | 12.5 |
| *Non-typeable* | 10 | 62.5 |
| **Colonization with *Staphylococcus aureus*** | 43 | 8.3 |
| **Negative for *Streptococcus pneumoniae* and *Staphylococcus aureus*** | 462 | 88.7 |
| **Received PCV13** | | |
| Yes | 2 | 0.4 |
| No | 519 | 99.6 |

NA-Not applicable; NP-Nasopharyngeal; OP- Oropharyngeal.

**Table 2. Associations between characteristics of participants and carriage of *Streptococcus pneumoniae*.**

| Characteristics | Tested | *Streptococcus pneumoniae* positive | Prevalence (95%CI) | P value |
|---|---|---|---|---|
| Age group | | | | |
| 50–59 | 70 | 2 | 2.86 (0.35–9.95) | 0.9355 |
| 60–69 | 165 | 4 | 2.42 (0.66–6.08) | |
| 70–79 | 197 | 7 | 3.55 (1.44–7.18) | |
| ≥80 | 89 | 3 | 3.37 (0.70–9.54) | |
| Gender | | | | |
| Male | 197 | 6 | 3.05 (1.13–6.52) | 0.9643 |
| Female | 324 | 10 | 3.09 (1.49–5.61) | |
| Month of study entry during 2021 | | | | |
| May | 190 | 5 | 2.63 (0.86–6.03) | <0.001 |
| June | 186 | 6 | 3.23 (1.20–6.89) | |
| July | 3 | 0 | 0.00 | |
| August | 12 | 0 | 0.00 | |
| September | 53 | 0 | 0.00 | |
| October | 74 | 2 | 2.70 (0.33–9.42) | |
| December | 3 | 3 | 100.00 | |
| Number of children aged 0–10 years in the family | | | | |
| 0 | 403 | 11 | 2.73 (1.37–4.83) | 0.5833 |
| 1–2 | 112 | 5 | 4.46 (1.46–10.11) | |
| ≥3 | 6 | 0 | 0.00 | |
| Smoker in the household | | | | |
| Yes | 78 | 4 | 5.13 (1.42–12.62) | 0.2535 |
| No | 443 | 12 | 2.71 (1.41–4.69) | |

(Table 4). There were three multidrug resistant isolates: one 6C (non-vaccine serotype), and two among the non-typeable isolates.

## Discussion

To the best of our knowledge, this is the first study conducted in Serbia with the aim to evaluate pneumococcal carriage in the population of middle-aged and older adults. It is also one of the rare studies that assessed Spn carriage during the current pandemic COVID-19. In order to improve sensitivity of Spn carriage detection [3,29], both NP and OP samples were obtained, and culture positive samples were serotyped. We found low prevalence of Spn carriage, as only 16 out of 521 tested participants were colonized, and identified 6 serotypes of Spn all originating from NP swabs.

**Table 3. Characteristics of participants with positive results (*Streptococcus pneumoniae* serotypes and non-typeable *Streptococcus pneumoniae* isolates).**

| Characteristics | *Streptococcus pneumoniae* serotypes | | | | | | Non-typeable *Streptococcus pneumonia* isolates | | | | | | | | | |
|---|---|---|---|---|---|---|---|---|---|---|---|---|---|---|---|---|
| | Participant No 1 | Participant No 2 | Participant No 3 | Participant No 4 | Participant No 5 | Participant No 6 | Participant No 1 | Participant No 2 | Participant No 3 | Participant No 4 | Participant No 5 | Participant No 6 | Participant No 7 | Participant No 8 | Participant No 9 | Participant No 10 |
| Age | 72 | 75 | 65 | 81 | 74 | 81 | 73 | 79 | 76 | 65 | 80 | 75 | 66 | 69 | 50 | 57 |
| Gender | Male | Female | Female | Male | Male | Female | Female | Female | Female | Female | Female | Female | Male | Male | Female | Male |
| Month of study entry during 2021 | May | May | May | June | June | October | May | May | June | June | June | June | October | December | December | December |
| Number of children aged 0–10 years in the family | 1 | 0 | 2 | 0 | 0 | 2 | 0 | 0 | 0 | 0 | 0 | 0 | 0 | 1 | 0 | 1 |
| Smoker in the household | Yes | No | No | No | Yes | No | Yes | Yes | No | No | No | No | No | No | No | No |
| Swab positive for *Streptococcus pneumoniae* | NP | NP | NP | NP | NP | NP | OP | NP | OP | NP | NP | NP | NP | NP | NP | NP |
| *Streptococcus pneumoniae* serotype | 35F | 15B | 6A | 11A | 18 C | 6C | NA | NA | NA | NA | NA | NA | NA | NA | NA | NA |

NP-Nasopharyngeal; OP- Oropharyngeal; NA-Not applicable.

**Table 4. Resistance patterns of *Streptococcus pneumoniae* isolates.**

| Antimicrobial agents | 6A (N = 1) | 6C (N = 1) | 15B/15C (N = 1) | 11A/11D (N = 1) | 35F (N = 1) | 18C (N = 1) | Non-typeable (N = 10) | Total (N = 16) (%) |
|---|---|---|---|---|---|---|---|---|
| Penicillin | | | | | | | 1 | 1 (6.25) |
| Ampicillin | | | | | | | 1 | 1 (6.25) |
| Amoxicillin (per os) | | | | | | | 1 | 1 (6.25) |
| Amoxicillin (i.v.) | | | | | | | 1 | 1 (6.25) |
| Amoxicillin-clavulanic acid (per os) | | | | | | | 1 | 1 (6.25) |
| Amoxicillin-clavulanic acid (i.v.) | | | | | | | 1 | 1 (6.25) |
| Ceftriaxone | | | | | | | | 0 (-) |
| Erythromycin | 1 | 1 | | | | | 8 | 10 (62.50) |
| Trimethoprim/sulfamethoxazole | | | | | | | | 0 (-) |
| Clindamycin | | 1 | | | | | | 1 (6.25) |
| Levofloxacin | | | | | | | | 0 (-) |
| Tetracycline | | 1 | | | 1 | | 5 | 7 (43.75) |

Several previous studies in different geographical areas showed different prevalence of Spn carriage in children, but there was only limited information on colonization among adults [2,4,5,7,8,29]. Previously observed extreme differences in prevalence of Spn carriage among adults and children (3.7% vs. 54%) in community and family settings, were explained by the presence of antibodies to Spn in adults or by a decreased number of receptors in the nasopharyngeal epithelium in adults [7,30].

The dynamics of the adult carrier state remained poorly investigated. In light of this, only a few longitudinal studies were available and mainly in families or in populations living in crowded conditions [28,31,32]. Despite all circumstances, culture-based methods estimated a low (1–10%) Spn colonization prevalence among adults in high-income countries [29]. Importantly, use of real-time PCR (qPCR) indicated that Spn colonization among adults is higher compared to prevalence obtained by culture-based methods [33,34].

Taking into account that our research was conducted during 2021, when two-epidemic waves of COVID-19 were recorded in our country, we found that the prevalence of Spn colonization among subjects ≥50 years of age was 3.1%. Similar prevalence (3.8%) was found in an Israel study conducted among 1300 screened adults [7], while prevalence in our study was higher than prevalence of 1.5%, obtained in Slovenian nursing home residents [35], or prevalence of 1.9% obtained from the United States study among retired community residents [9], or prevalence of 2.2%, among outpatients ≥60 years of age in Brazil [36], and the prevalence of 2.3% found after testing adults above 60 years of age in Portugal [3]. We are aware that our results are not comparable with the prevalence of aforementioned studies, but it is important to notice that we performed a sample size calculation for a study estimating a population

prevalence. In addition, we found that prevalence of *S. aureus* carriers was 8.3% which possibly influenced Spn carriage in our participants as an additional reduction factor, probably due to the competition between Spn and *S. aureus* for the niche of the nasopharynx [36].

According to our results, most of the Spn isolates were non-typeable, which probably means that they did not have the polysaccharide capsule [35]. Among serotyped Spn isolates, we found four vaccine (6A, 11A, 15B, and 18C) serotypes, and only two (6C and 35F) non-vaccine ones. Vaccine serotypes (6A and 18C) are covered by the PCV13 vaccine. Vaccine serotype 18C along with other two other (11A and 15B) is also present in the PPSV23 vaccine, while serotype 6A is only present in PCV13 vaccine. Referring to the results of above-mentioned studies, serotype 6A was the most prevalent serotype in the Israeli study [7], serotype 6B in the Slovenian study [35], serotype 3 in the United States study [9], serotypes 3 and 6C in the Brazilian study [36], and serotype 19A in the study from Portugal [3]. As it is well established, the serotype distribution of Spn in some population could be affected by the reception of vaccine since pneumococcal vaccines can prevent nasopharyngeal colonization [35,37], but the fact that only two participants in our study were vaccinated precludes any conclusion regarding the effect of vaccination on Spn carriage in our population.

Although the Spn carriage in our study was low, this population (≥50 years of age) has a potential high-risk for IPD [38,39]. As we previously noted, seasonality also affects pneumococcal carriage rates, and therefore prevalence of IPD. The pneumococci are transmitted better during cooler and drier months [6]. It is possible that low prevalence of Spn carriage in our study was also affected by the fact that our samples were mainly collected during the May and June i.e. during "off season" for detection of pneumococcus. In addition, it is also a known that influenza viruses have a facilitatory effect on pneumococcus, by increasing acquisition, bacterial load, transmission, or disease severity [40]. In line with this, during 2021, there were no influenza confirmed cases in our country, and viral activity was low throughout the year and therefore these effects were absent.

Regarding the antimicrobial resistance pattern analysis among 16 positive isolates, we found that most Spn isolates were resistant to erythromycin (62.5%), and tetracycline (43.8%). Regarding serotypes, one 6A and one 6C as well as eight non-typeable isolates showed resistance to erythromycin, and by one of 6C and 35F and five non-typeable ones showed resistance to tetracycline. In addition, one non-typeable isolate displayed resistance to penicillin, ampicillin, and amoxicillin/amoxicillin-clavulanic acid. Similar results were widely published before [7,11,41].

The main advantage of this study is that it was the first study that provided the baseline measurement addressing the issue of Spn carriage, serotype identification as well as antimicrobial resistance patterns among middle-aged and older adults in our country. Moreover, it adds new data on Spn carriage during pandemic COVID-19 that may serve others as well. Despite the low prevalence of Spn carriage, we believe that the results of this study are comparable with those from other studies performed elsewhere [3,7,9,35,36].

There are also several limitations of this study. First, this study was conducted in a single city, from a highly urbanised area; therefore, there is a need for larger, multi-centre longitudinal studies, across the country to obtain a more representative view of the pneumococcal epidemiology. Second, due to a small number of positive cases (probably due to the fact that we did not tested all samples by PCR technique but only those which previously were culture positive), we were unable to provide analyses of previously recognized associated risk factors of Spn carriage [3,9,10,13,29–31] in our study population. Further study with a larger sample size and wide inclusion criteria are necessary to address this topic. Third, although we excluded all study subjects who used antibiotics two weeks prior sampling, we did not predict to separate sampling among healthy vs. sick (acute upper respiratory infection) participants. Fourth, due

to exclusion of the participants with comorbidity or confirmed immunodeficiency, we have possibly omitted some colonized individuals and consequently obtained smaller prevalence of Spn carriage, and therefore we were unable to determine true prevalence of Spn colonization among middle-aged and older adults population with comorbidities. Fifth, due to COVID-19 pandemic and occasional interruptions of research (study was conducted during COVID-19 pandemic, and non-pharmaceutical intervention targeting the SARS-CoV-2 virus transmission could further reduce transmission and carriage of pneumococci), there was no possibility to assess a potential seasonal fluctuation, and incidence rates of Spn colonisation in our city. However, an inverse trend between the numbers of COVID-19 cases and Spn positive samples among patients ≥50 years of age has been noticed in our study. According to the data of the IPHV [42], a majority of COVID-19 cases in Novi Sad were registered during two months (March and October 2021) when Spn carriage was rarely detected (only two cases in October). On the other hand, ¾ of all Spn carriage were registered during May and June 2021, when only sporadic COVID-19 cases were detected in our city. It is also possible that, a similar picture with low number of COVID-19 cases and high number Spn colonized subjects could have occurred during July and August 2021 as in May and June 2021, but the research was temporarily interrupted in this period due to holidays. Despite these indications, it remains unclear whether the raising number of COVID-19 cases could have suppressed the risk of Spn carriers among the middle-aged and older adults in Novi Sad. Sixth, due to aforementioned reasons, our study was not performed in the coolest and driest months of the year, which might have influenced our results. Finally, we did not predict the sampling from the children and grandparents within the families of our participants. Recently published study showed that living with children increased the risk of Spn acquisition among healthy adults up to 10-fold [29]. Thus, further research could provide the estimation of concordance by Spn serotypes distribution in children and their older family members and potentially determine source of horizontal Spn carriage transmission.

In conclusion, data from this study performed for the first time in our territory, suggests that Spn carriage in middle-aged and older adults is low, but it is worth mentioning that some of the isolated serotypes (namely 6A, 11A, 15B, and 18C) are covered by either PCV13 or PPSV23 vaccines against pneumococcus that are both available in Serbia. Although we were unable to give conclusions regarding the potential effect of pneumococcal vaccines in the prevalence and dynamics of Spn carriage, our results support the need to increase the coverage of vaccination against Spn in middle-aged and older adults, which is very low in our country. High antimicrobial resistance against erythromycin and tetracycline, and low against all others tested antibiotics including penicillin was also observed. Importantly, despite its limitations and a limited number of Spn isolates, this study could be used as the basis for future more comprehensive investigations of pneumococci serotypes and drivers for antibiotic resistance, not only in Serbia, but in a similar study population, as well.

## Supporting information

**S1 Appendix. Information leaflet for subjects.**
(DOCX)

**S2 Appendix. Informed consent of the subject.**
(DOCX)

**S3 Appendix. Survey questionnaire.**
(DOCX)

## Acknowledgments

We thank the physicians from Health Centre Novi Sad (dr Ana Miljković and dr Tatjana Usorac) as well as the clinical laboratory technicians at the Institute of Public Health of Vojvodina, Novi Sad who participated in this project.

## Author Contributions

**Conceptualization:** Mioljub Ristić, Milan Djilas, Miloš Marković.

**Data curation:** Mioljub Ristić, Biljana Milosavljević, Milan Djilas.

**Formal analysis:** Mioljub Ristić, Biljana Milosavljević, Milan Djilas, Miloš Marković.

**Investigation:** Mioljub Ristić.

**Methodology:** Mioljub Ristić, Biljana Milosavljević, Milan Djilas.

**Project administration:** Mioljub Ristić.

**Supervision:** Vladimir Petrović, Milan Djilas, Miloš Marković.

**Validation:** Vladimir Petrović, Mioljub Ristić, Biljana Milosavljević, Miloš Marković.

**Visualization:** Mioljub Ristić, Miloš Marković.

**Writing – original draft:** Vladimir Petrović, Mioljub Ristić, Miloš Marković.

**Writing – review & editing:** Vladimir Petrović, Miloš Marković.

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
