## [Decision Letter · Decision Letter 0]

15 Mar 2022

PONE-D-22-02347Pneumococcal nasopharyngeal carriage in adults aged 50 years and older in outpatient health care facility during pandemic COVID-19 in Novi Sad, SerbiaPLOS ONE

Dear Dr. Ristić,

Thank you for submitting your manuscript to PLOS ONE. After careful consideration, we feel that it has merit but does not fully meet PLOS ONE’s publication criteria as it currently stands. Therefore, we invite you to submit a revised version of the manuscript that addresses the points raised during the review process.

The manuscript has been assessed by three reviewers; the comments are available below. The reviewers have raised a number of concerns about the concepts, methodology, clarity in presentation of the work and the data, they recommend rejection of the manuscript or revisions to improve the clarity in presentation and writing and to provide a fuller outline of the methodology, main results and discussion.

I agree with the reviewers and I highlight one of the major problems (mentioned by the three reviewers) of the manuscript as it stands: The lack of molecular methods for the detection of carriage.

Please carefully revise the manuscript to address all the points raised by the three reviewers. Please go back to the samples and study them with molecular methods to improve detection rate of carriage. Please submit a text that clearly and definitely addresses all these issues, otherwise the manuscript will not be acceptable for publication in PLoS One.

We look forward to receiving your revised manuscript.

Kind regards,

Jose Melo-Cristino, M.D., Ph.D.

Academic Editor

PLOS ONE

Journal Requirements:

Reviewers' comments:

Reviewer's Responses to Questions

**Comments to the Author**

1. Is the manuscript technically sound, and do the data support the conclusions?

Reviewer #1: No

Reviewer #2: Partly

Reviewer #3: Yes

2. Has the statistical analysis been performed appropriately and rigorously? 

Reviewer #1: No

Reviewer #2: No

Reviewer #3: Yes

3. Have the authors made all data underlying the findings in their manuscript fully available?

Reviewer #1: Yes

Reviewer #2: No

Reviewer #3: Yes

4. Is the manuscript presented in an intelligible fashion and written in standard English?

Reviewer #1: Yes

Reviewer #2: No

Reviewer #3: No

5. Review Comments to the Author

Reviewer #1: The study by Ristic and co-workers aimed to determine the prevalence of pneumococcal carriage in adults over 50 years of age in an urban region of Serbi, and to establish a baseline measurement of antimicrobial susceptibility patterns and (presumably) risk factors for pneumococcal colonization.

This was a prospective study conducted during the Covid pandemic, from May to December 2021, and included individuals visiting their physicians in the Primary Health Care Center. One nasopharyngeal and one oropharyngeal sample was collected from each participant.

The study reports a pneumococcal carriage prevalence of around 3%, which is in agreement with reports from other settings in the same age group, when using culture-based methods. Nevertheless, given the main objectives established for this study, it suffers from a severe limitation in what concerns the number of pneumococcal positive samples, which precludes that any meaningful conclusions can be taken from the data.

The study reports concern in calculation of the sample size. Nevertheless, it is not mentioned which expected prevalence was established for that calculation and whether the results obtained differed significantly from the ones expected (which could account for the low number of pneumococcal positive samples).

Minor comments that could improve the manuscript:

Line 48: “siblings” should be replaced with “contact with children”. This should be revised in the entire manuscript as the word sibling means a brother or sister.

Line 51: the sentence starting with “vaccine serotypes” should be removed

Line 120: Please provide more detailed information on the sample size calculation (estimated prevalence, degree of confidence, etc)

Line 149-151: Please provide information on whether OP samples were processed in the same manner. Please explain the rational for not using a selective medium for isolation of pneumococci

Tables 2 and 3 should be combined into one as they are reporting the same type of results.

Figure S1 is impossible to analyze.

Lines 254-255: PMID: 32433504 reference should be included to further support the sentence.

Reviewer #2: In the manuscript submitted to PLoS One DrRustic and his colleagues report on the prospective study on pneumococcal carriage conducted in community-dwelling adults aged 50 to 93 years in the city of Novi Sad, Serbia.

The objective of the study was “to obtain a baseline measurement of S. pneumoniae carriage,antibiotic susceptibility pattern, and associated (risk?) factors” that could “serve as additional evidence in order to increase (pneumococcal vaccine?) immunization coverage among adults and elderly individuals”.

For that, study team collected paired nasopharyngeal and oropharyngeal samples from 521 adults over the period of seven months of 2021. Nasopharyngeal samples were culture-enriched for pneumococcus prior to being tested for S. pneumoniae with conventional diagnostic culture. It resulted in isolation of S. pneumoniae from 14 nasopharyngeal samples, including six (geno)serotypeable and eight non-typeable pneumococci. It is unclear how oropharyngeal samples were tested for S. pneumoniae, but authors report on two more individuals identified as carriers of non-typeable pneumococci based on culture-positive oropharyngeal swab. It translates into overall S. pneumoniae carriage prevalence of 3.1% and of (geno)serotypeable pneumococci of 1.2%, and represents one of the lowest carriage rates ever reported for adults that age (Arguedas et al, DOI: 10.1080/14760584.2020.1750378).

The reviewer agrees with authors that such a low prevalence could be expected considering the vast majority of samples have been collected outside the acute viral respiratory infections season.

Also, since the study was conducted during COVID-19 pandemic, non-pharmaceutical intervention targeting the SARS-CoV-2 virus transmission could further reduce transmission and carriage of pneumococci.  

However, low prevalence of carriage can be also explained by low sensitivity of the method used. There is evidence that the conventional diagnostic culture of nasopharyngeal swab allows detection of only a fraction of carriers among adults (see Arguedas et al.), with the conventional diagnostic culture of oropharyngeal samples being even less sensitive.  Accurate detection of S. pneumoniae in upper airways of adults apparently requires application of both, conventional culture and molecular diagnostics that are most sensitive when applied combined to oropharyngeal samples, as described in the reference #30 and also in DOI: 10.1093/infdis/jiaa558, and in DOI: 10.1038/s41598-020-65399-x). Nevertheless, the methods applied to detect carriage in this study are actually in line with the latest WHO recommendations (doi:10.1016/j.vaccine.2013.08.062). Hence, reviewer’s comment should not be considered to represent the criticism.

Major points to address:

1. The manuscript would benefit from thorough editing for scientific language. Particularly poorly written is the Results section. It lacks focus.  

2. There is no statistical analysis of data, or any result reported in the study. It is likely that the low number of carriage events detected precludes differences to be significant, but statistics would help to avoid reporting on false association. For example, contrary to the text in line 213, the difference between prevalence of NT carriage among men and women (3 of 197 or 1,5% versus 7 of 324 or 2.2%) was not significant (p=0.75). Also, instead of reporting on age of individual carriers it would be more appropriate to test if the distribution of age was any different between carriers and non-carriers. Lacks of statistical analysis is the clear weakness of the manuscript 

3. Can the authors elaborate if the study objective was reached? How the information on low rate of carriage detected in adults in Novi Sad may influence the future of pneumococcal vaccination in Serbia? 

4. Authors report (line 170) four ATCC strains to be used as control in smPCRs that has bee applied to determine serotype of cultured S. pneumoniae isolates. Were there any positive controls for serotypes other than the capsular polysaccharide types expressed by these four ATCC strains?  

Minor points:

One of two study endpoints (line 149) was carriage of Staphylococcus aureus. Authors report (Table 1) on 43 individuals identified as S. aureus carriers. Relevance of that finding remains obscure. It does not seem pneumococcus and S. aureus were detected simultaneously in any participant. It may imply a negative interference between species. 

In the discussion (line 248-249) authors attribute the low rate of carriage to age-related changes in the physiology of nasopharynx and immune responses. Regev-Yochay et al cited by authors do not report on any such changes. Also, since the immunosenescence is associated with increased sensitivity to pneumococcal disease, it can be argued that the sensitivity to pneumococcal carriage supposed to increases in older age.   

It is reported (line105) that not only PCV13 but PPS23 was used in Serbia, and (line 139) the informationon PPS23 vaccination was recorded in the study. There is no information on PPS23 vaccination status in the manuscript. Was none of the participants vaccinated with PPS23?  

Section starting in line 219: Was any of the resistance phenotypes over-represented among non-typeable isolates?

Except for penicillin and ceftriaxone, the information about susceptibility/resistance to any other beta-lactam(s) seems irrelevant. Also,considering low virulence of NT pneumococci, is information about i.v. formulas of a relevance?

Authors reports (line 245) on positive nasopharyngeal or oropharyngeal samples to be serotyped in the study. It seems that S. pneumoniae isolates cultured from samples, and not nucleic acids extracted from the polymicrobial samples were tested in smPCR. Rephrase? 

Describing children living in participants' household as "siblings" is misleading as it implies study subjects were actually brothers or sisters of these children. Also, single-child families cannot be described as families with siblings. 

It seems appropriate to describe the study population as “middle-aged and older adults” instead of “adults and elderly adults”. For one, older adults are a subcategory within adults. 

Line 306: Since serotype 6A is unique for PCV13 and serotypes 11A and 15B for PPS23, should “covered by both” be replaced with “covered by either”?

Line 129: replace “signs” with “symptoms”? 

It might be appropriate to consistently use the same term ‘participants’ to describe subjects sampled in the study: replace ‘respondents’in line 212? Table 2 and 3, it does not seem necessary to report on a feature that is universal for all isolates: on vaccination status of carriers inboth tables, and on all isolates listed in Table 2 being cultured from nasopharyngeal samples.

Reviewer #3: In this manuscript the authors present a study on the pneumococcal carriage from NP and OP in adults ≥ 50 years during COVID-19 in Novi Sad, Serbia. The topic is interesting and of importance as, in contrast to children, there are not many studies in adults. However, there are some issues that need to be dealt with.

Points to be addressed:

1. Authors reported carriage data obtained only by culture-based methods (16 positive samples/1042). Why did they not screen all NP/OP samples for pneumococcal presence using molecular methods as well? It is now agreed that carriage studies should be performed by merging the classical culture-based method with the very sensitive molecular method of qPCR and testing with qPCR the DNA extracted from samples culture-enriched for pneumococci.In addition, it is recommended to include more than one specific target for pneumococcal identification by qPCR, such as lytA, piaB or SP2020, to avoid false positivity due to other viridans streptococci. It is likely that by using molecular methods, carriage rate estimates would be higher. On the other hand, authors performed S. pneumoniae confirmation by lytA and molecular serotyping on isolated streptococci, therefore they can manage this procedure.

2. Authors should describe the Sample Processing (page 5, line 143) in a more detailed way since they report data for both NP and OP but in some cases they refer only to NP samples (see line 149). Since this study is based on NP and OP analysis and 2 positive samples were from OP, authors should use the term nasopharyngeal only if associated with the source of sampling. Accordingly, Authors should refer to “pneumococcal carriage” and not to “nasopharyngeal carriage” throughout the text, tables and in the title of the manuscript.

3. In DNA extraction (page 5 line 156) the overnight growth agar plate was that from the enrichment step or was referred to the cultured strain? Please specify.

4. As per the Serotyping Identification, authors screened for all known serotypes by using Quellung and molecular tests? If no, they should report samples with no identified serotype as Not Determined and not as Non-typeable. Further, in Discussion authors speculate that non-typeable isolates were likely to miss the polysaccharide capsule (lines 264-265), are they sure?

5. In Results which is the importance to describe the characteristics of participants associated with serotyped and not serotyped S. pneumoniae? It is not clear to me.

6. In Results the description of antimicrobial susceptibility pattern is quite confuse, authors report alternatively antibiotic susceptibility rate and antibiotic resistance rate. It should be better to report only the antibiotic resistance rate for each specific class of antibiotics. The same should be corrected in the Abstract.

7. In Results the paragraph related to distribution of Covid-19 and follow-up of participants should be reported only as a discussion point, as a limit of the study.

6. PLOS authors have the option to publish the peer review history of their article (what does this mean?). If published, this will include your full peer review and any attached files.

Reviewer #1: No

Reviewer #2: No

Reviewer #3: No

---

## [Author Response · Author response to Decision Letter 0]

17 Jun 2022

PONE-D-22-02347

Pneumococcal nasopharyngeal carriage in adults aged 50 years and older in outpatient health care facility during pandemic COVID-19 in Novi Sad, Serbia

PLOS ONE 

Dear Dr. Ristić,

Thank you for submitting your manuscript to PLOS ONE. After careful consideration, we feel that it has merit but does not fully meet PLOS ONE’s publication criteria as it currently stands. Therefore, we invite you to submit a revised version of the manuscript that addresses the points raised during the review process.

The manuscript has been assessed by three reviewers; the comments are available below. The reviewers have raised a number of concerns about the concepts, methodology, clarity in presentation of the work and the data, they recommend rejection of the manuscript or revisions to improve the clarity in presentation and writing and to provide a fuller outline of the methodology, main results and discussion.

Editor: I agree with the reviewers and I highlight one of the major problems (mentioned by the three reviewers) of the manuscript as it stands: The lack of molecular methods for the detection of carriage.

Please carefully revise the manuscript to address all the points raised by the three reviewers. Please go back to the samples and study them with molecular methods to improve detection rate of carriage. Please submit a text that clearly and definitely addresses all these issues, otherwise the manuscript will not be acceptable for publication in PLoS One.

- Dear Editor, thank you for these suggestions. We are fully aware of the limitations of our study, and we have listed them and commented at the end of Discussion section of our revised manuscript. On the other hand, we do believe that our study has some merrit and add some important additional information that can be of interest for the wider scientific community and we elaborate on that in more details in response to specific concerns and suggestions raised by all three Reviewers. However, here we would like to put the emphasis on some major points and (dis)advantages of our study. We think that the main advantage of this study is in the fact that we presented the results on the pneumococcal carriage from nasopharynx and oropharynx in adults ≥ 50 years of age in Novi Sad, Serbia, which was done for the first time in our country, but also represents one of the few studies that assessed penumococcal carriage during COVID-19 pandemic. Despite all limitations, we believe that this topic is interesting and of importance mainly as, in contrast to children, there are not many studies performed in adults, especially not in a situation where socail distancing and non-pharmaceutical intervention targeting the SARS-CoV-2 virus transmission could further reduce transmission and carriage of pneumococci. As we highlighted in the main text of the manuscript, methodology of our study is based on propositions widely used in various studies, as seen from literature sources (see ref. 21-27), but, of special note, it is in accordance with the recommendations of World Health Organization Pneumococcal Carriage Working Group (ref. 18), CDC (ref. 20), and the European Committee on Antimicrobial Susceptibility Testing (ref. 28). As reviewer #2 underlined: „Nevertheless, the methods applied to detect carriage in this study are actually in line with the latest WHO recommendations“. He also pointed out that his comment on methodology used should not be considered to represent the criticism. Even so, we are aware that we could re-test our samples using PCR, and eventually identify some additional carriers and serotypes. Unfortunately, we could not tested the samples again, simply because they were not preserved. Considering this, we stated in the manuscript that the limitation of this study was the abscence of molecular (PCR) testing of all collected samples that could have led to a lower prevalence of Spn positive samples among our study subjects. 

We look forward to receiving your revised manuscript.

Kind regards,

Jose Melo-Cristino, M.D., Ph.D.

Academic Editor

PLOS ONE

Journal Requirements:

We have now corrected grant information in “Funding Information” and “Fianancil Disclosure” so that they match.

Reviewers' comments:

Reviewer's Responses to Questions

Comments to the Author

1. Is the manuscript technically sound, and do the data support the conclusions?

Reviewer #1: No

Reviewer #2: Partly

Reviewer #3: Yes

2. Has the statistical analysis been performed appropriately and rigorously?

Reviewer #1: No

Reviewer #2: No

Reviewer #3: Yes

3. Have the authors made all data underlying the findings in their manuscript fully available?

Reviewer #1: Yes

Reviewer #2: No

Reviewer #3: Yes

4. Is the manuscript presented in an intelligible fashion and written in standard English?

Reviewer #1: Yes

Reviewer #2: No

Reviewer #3: No

5. Review Comments to the Author

Response to Reviewers

IMPORTANT!

Comments to the Authors are in regular fonts, while the comments from us to the academic Editor and Reviewers are in the italic fonts.

Reviewer #1: 

The study by Ristic and co-workers aimed to determine the prevalence of pneumococcal carriage in adults over 50 years of age in an urban region of Serbi, and to establish a baseline measurement of antimicrobial susceptibility patterns and (presumably) risk factors for pneumococcal colonization.

This was a prospective study conducted during the Covid pandemic, from May to December 2021, and included individuals visiting their physicians in the Primary Health Care Center. One nasopharyngeal and one oropharyngeal sample was collected from each participant.

The study reports a pneumococcal carriage prevalence of around 3%, which is in agreement with reports from other settings in the same age group, when using culture-based methods. Nevertheless, given the main objectives established for this study, it suffers from a severe limitation in what concerns the number of pneumococcal positive samples, which precludes that any meaningful conclusions can be taken from the data.

- We agree that the number of pneumococcal positive samples was rather low, but, as the Reviewer has already noted, the prevalence of Spn colonization among subjects ≥50 years in our study was comparable with those obtained in similar studies using culture-based methods from other countries. Taking into account the scarcity of available data concerning pneumoccocal carriage in adults, we do believe that, despite limited number of positive samples and pneumococcal isolates, the results of this study are of importance and can be compared with data from other studies on the same topic. Moreover, this is one of the rare studies that assessed pneumococcal cariage during COVID-19 pandemic during which strict epidemiological measures and social distancing have ceratinly interfered with the circulation of other pathogens (including Spn) and affected their transmission and epidemiology in general in population across all ages. Therefore, we think that this study can add some additional information that can be of interest for the wider spectrum of reaserchers.

The study reports concern in calculation of the sample size. Nevertheless, it is not mentioned which expected prevalence was established for that calculation and whether the results obtained differed significantly from the ones expected (which could account for the low number of pneumococcal positive samples).

- Thank you very much for this suggestion. So, in accordance with the sample size calculation for a study estimating a population prevalence (ref. Naing L, Winn T, Rusli BN. Practical Issues in Calculating the Sample Size for Prevalence Studies. Archives of Orofacial Sciences 2006; 1: 9-14), we planned to include between 350 and 500 subjects in the study and we presumed to detect pneumococci (PCR positive for Spn) in the samples from up to 5% of tested subjects. In addition, since the total number of adults ≥50 years of age in Novi Sad was estimated to be around 120.000, we expected and managed to include in the study a number of subjects that corresponds to 0.4% of the total targeted population which seemed as appropriate size of the sample. For the sake of clarity, this was explained in more details in the Materials and methods section (Study Site, Design, and Participants) of the revised version of the manuscript.

Minor comments that could improve the manuscript:

Line 48: “siblings” should be replaced with “contact with children”. This should be revised in the entire manuscript as the word sibling means a brother or sister.

- Thank you for this comment. We replaced this term with the proposed expression throughout the entire paper.

Line 51: the sentence starting with “vaccine serotypes” should be removed

- We absolutely agree with this and removed this sentence from the present version of the manuscript.

Line 120: Please provide more detailed information on the sample size calculation (estimated prevalence, degree of confidence, etc)

- As previously mentioned, we added more detailed information on the sample size calculation in the Materials and methods section (Study Site, Design, and Participants subsection) of the revised version of the paper.

Line 149-151: Please provide information on whether OP samples were processed in the same manner. Please explain the rational for not using a selective medium for isolation of pneumococci.

- Thank you for this request that will further explain the methodology used in this study. We processed both NP and OP in the same manner, according to WHO Pneumococcal Carriage Working Group the standard operating procedures. In fact, selective medium was used for the isolation of pneumococci – All samples were cultured within 6 hours on Columbia agar supplemented with 5 % sheep blood and gentamycin 5 mg/l. Plates were inoculated using the original swab from Amies medium (mixed thoroughly using vortex) and incubated overnight at at 35-37°C in 5-10% CO2 for 24 h. In order to avoid misunderstanding, this explanation was added in the appropriate section of the paper (Materials and methods/Sample processing).

Tables 2 and 3 should be combined into one as they are reporting the same type of results.

- Thank you for this helpful suggestion. We agree with this and we have merged data from Tables 2 and 3 into newly formatted table (Table 2 in the revised manuscript).

Figure S1 is impossible to analyze.

- Our aim was to refer to how the pandemic and a number of COVID-19 cases in the city of Novi Sad affected the level of positive samples in patients ≥50 years of age in our study. We do agree, that in the form that it was originally presented, Figure S1 was neither enough comprehensive nor easy to analyze. However, we did observe some inverse trend between the numbers of COVID-19 cases and Spn positive samples in various months during the study period, so we believe that this observation may be the basis for further research on the this topic. Nevertheless, following the Reviewer #1’s comment and in accordance with the suggestion of the Reviewer #3, we have decided to remove this section and related figure from the Results section, but referred to this issue and gave additional explanation related to COVID-19 and Spn carriage in the Limitation of the study at the end of the Discussion section of the present version of the manuscript.

Lines 254-255: PMID: 32433504 reference should be included to further support the sentence.

- We added the suggested reference (ref No 34 in the present version of the manuscript: Almeida ST, Pedro T, Paulo AC, de Lencastre H, Sá-Leão R. Re-evaluation of Streptococcus pneumoniae carriage in Portuguese elderly by qPCR increases carriage estimates and unveils an expanded pool of serotypes. Sci Rep. 2020; 10(1):8373. doi: 10.1038/s41598-020-65399-x. PMID: 32433504; PMCID: PMC7239868).

Reviewer #2:

In the manuscript submitted to PLoS One DrRustic and his colleagues report on the prospective study on pneumococcal carriage conducted in community-dwelling adults aged 50 to 93 years in the city of Novi Sad, Serbia.

The objective of the study was “to obtain a baseline measurement of S. pneumoniae carriage,antibiotic susceptibility pattern, and associated (risk?) factors” that could “serve as additional evidence in order to increase (pneumococcal vaccine?) immunization coverage among adults and elderly individuals”.

For that, study team collected paired nasopharyngeal and oropharyngeal samples from 521 adults over the period of seven months of 2021. Nasopharyngeal samples were culture-enriched for pneumococcus prior to being tested for S. pneumoniae with conventional diagnostic culture. It resulted in isolation of S. pneumoniae from 14 nasopharyngeal samples, including six (geno)serotypeable and eight non-typeable pneumococci. It is unclear how oropharyngeal samples were tested for S. pneumoniae, but authors report on two more individuals identified as carriers of non-typeable pneumococci based on culture-positive oropharyngeal swab. It translates into overall S. pneumoniae carriage prevalence of 3.1% and of (geno)serotypeable pneumococci of 1.2%, and represents one of the lowest carriage rates ever reported for adults that age (Arguedas et al, DOI: 10.1080/14760584.2020.1750378).

The reviewer agrees with authors that such a low prevalence could be expected considering the vast majority of samples have been collected outside the acute viral respiratory infections season.

Also, since the study was conducted during COVID-19 pandemic, non-pharmaceutical intervention targeting the SARS-CoV-2 virus transmission could further reduce transmission and carriage of pneumococci. 

However, low prevalence of carriage can be also explained by low sensitivity of the method used. There is evidence that the conventional diagnostic culture of nasopharyngeal swab allows detection of only a fraction of carriers among adults (see Arguedas et al.), with the conventional diagnostic culture of oropharyngeal samples being even less sensitive. Accurate detection of S. pneumoniae in upper airways of adults apparently requires application of both, conventional culture and molecular diagnostics that are most sensitive when applied combined to oropharyngeal samples, as described in the reference #30 and also in DOI: 10.1093/infdis/jiaa558, and in DOI: 10.1038/s41598-020-65399-x). Nevertheless, the methods applied to detect carriage in this study are actually in line with the latest WHO recommendations (doi:10.1016/j.vaccine.2013.08.062). Hence, reviewer’s comment should not be considered to represent the criticism.

Major points to address:

1. The manuscript would benefit from thorough editing for scientific language. Particularly poorly written is the Results section. It lacks focus. 

-Thank you for this helpful suggestion. A thorugh editing and English language spelling and grammar check were performed throughout the manuscript. The special emphasis was put on Results section, but some other parts were also considerably revised in order to make it more intelligible and easier to follow.

2. There is no statistical analysis of data, or any result reported in the study. It is likely that the low number of carriage events detected precludes differences to be significant, but statistics would help to avoid reporting on false association. For example, contrary to the text in line 213, the difference between prevalence of NT carriage among men and women (3 of 197 or 1,5% versus 7 of 324 or 2.2%) was not significant (p=0.75). Also, instead of reporting on age of individual carriers it would be more appropriate to test if the distribution of age was any different between carriers and non-carriers. Lacks of statistical analysis is the clear weakness of the manuscript.

-Thank you for this suggestion. In accordance with abovementioned comments, we restructured results section of the manuscript and included statistical analysis in it. As a result, a new table on associations between characteristics of participants and carriage of Streptococcus pneumoniae was added (Table 2 in the present version of the manuscript).

3. Can the authors elaborate if the study objective was reached? How the information on low rate of carriage detected in adults in Novi Sad may influence the future of pneumococcal vaccination in Serbia? 

-Thank you for this suggestion. As we stated in the Introduction section “The objective of this study was to obtain a baseline measurement of Spn carriage, antibiotic susceptibility pattern, and associated factors of Spn among subjects ≥50 years of age in the City of Novi Sad”, In other words, our aim was to obtain the first estimation of the prevalence in our population, which could lay the foundations for further research on the same topic in a wider setting that may encompass the whole country. In addition, although the Spn carriage in adults and the elderly was found to be low, four isolated serotypes (6A, 11A, 15B, and 18C) were covered by both PCV13 or PPSV23 vaccines against pneumococcus that are available in Serbia, but are rarely adminstered to elderly. So, it would be reasonable to assume that many of colonized adults may harbour vaccinal serotypes, suggesting that there is a need to increase the coverage of vaccination against Spn in adults and the elderly. Thus, we think that the results of this study, despite several limitations, including limited number of isolates, could be used as the basis for future more comprehensive investigations of pneumococci serotypes and drivers for antibiotic resistance, not only in Serbia, but in a similar study population, as well.

4. Authors report (line 170) four ATCC strains to be used as control in smPCRs that has bee applied to determine serotype of cultured S. pneumoniae isolates. Were there any positive controls for serotypes other than the capsular polysaccharide types expressed by these four ATCC strains? 

-Thank you for this question. Our answer is no, since only those four strains were available to us as the positive controls. More precisely, ATCC6305 strain was positive for serotype 5, ATCC49619 strain was positive for serotype 19F, ATCC6303 strain was positive for serotype 3, and ATCC700677 strain was positive for serotype 14.

Minor points:

One of two study endpoints (line 149) was carriage of Staphylococcus aureus. Authors report (Table 1) on 43 individuals identified as S. aureus carriers. Relevance of that finding remains obscure. It does not seem pneumococcus and S. aureus were detected simultaneously in any participant. It may imply a negative interference between species. 

-You are absolutely right. There were no participants who had both Spn and S. aureus at the same time. Altough determination of the carriage of the S. aureus was not the main goal of this study, we wanted to present this data as an additional finding, similar to the results of another study (ref. No 36: Zanella RC, Brandileone MCC, Almeida SCG, de Lemos APS, Sacchi CT, Gonçalves CR, et al. Nasopharyngeal carriage of Streptococcus pneumoniae, Haemophilus influenzae, and Staphylococcus aureus in a Brazilian elderly cohort. PLoS One. 2019; 14(8):e0221525. doi: 10.1371/journal.pone.0221525. PMID: 31437226; PMCID: PMC6705818) in which they included subjects with mean age of 81.9 years and found a prevalence of S. aureus to be much higher than the one of Spn (15.9% and 2.2%, respectively). In our study, participants were younger (the mean age was 70.2 years), and the prevalence of S. aureus was lower (8.3%), while the prevalence of Spn was higher (3.1%) than in the abovementioned study. In addition, in this study from Zanella RC et colleagues, among 820 included participants, co-colonization of S. pneumonia, S. aureus and/or H. influenzae was identified in only 9 participants (1.1%) at visit 1, and five additional participants at visit 2. The authors of this study presumed that thier results are consistent with the reports on a co-habitation relationship of these three bacteria in the respiratory mucosa and the competition between these bacteria and S. aureus for the niche of the nasopharynx. In our study, we included 521 participants in total, but contrary to the previous one where some of participants had chronic disease, we excluded participants with a confirmed immunodeficiency, which could explain the lower level of S. aures carriage and a lack of co-colonization among our study subjects. We referred to this issue and cited corresponding study in the appropriate part of the Discussion section of the revised version of our manuscript.

In the discussion (line 248-249) authors attribute the low rate of carriage to age-related changes in the physiology of nasopharynx and immune responses. Regev-Yochay et al cited by authors do not report on any such changes. Also, since the immunosenescence is associated with increased sensitivity to pneumococcal disease, it can be argued that the sensitivity to pneumococcal carriage supposed to increases in older age.

- We agree with your suggestions. However, we referred to a statement from the cited study from Regev-Yochay et al. on this issue: “Optional explanations could be the presence of antibodies to S. pneumoniae in adults (boosted by the presence of S. pneumoniae in their children) or a decreased number of receptors in the nasopharyngeal epithelium”. In order to further support this notion, we added another reference (ref. 30. Simell B, Korkeila M, Pursiainen H, Kilpi TM, Käyhty H. Pneumococcal carriage and otitis media induce salivary antibodies to pneumococcal surface adhesin a, pneumolysin, and pneumococcal surface protein a in children. J Infect Dis. 2001;183(6):887-96. doi: 10.1086/319246. Epub 2001 Feb 21. PMID: 11237805) in the revised version of the manuscript.

It is reported (line105) that not only PCV13 but PPS23 was used in Serbia, and (line 139) the informationon PPS23 vaccination was recorded in the study. There is no information on PPS23 vaccination status in the manuscript. Was none of the participants vaccinated with PPS23?

- We kept the same text at lines 105 and 139, but added additional explanation in the first paragraph of the Results section (Socio-demographic characteristics) explaining that there were only two participants who received PCV13 vaccines and who were Spn negative, while none of the participants in our study had previously received PPSV23.

Section starting in line 219: Was any of the resistance phenotypes over-represented among non-typeable isolates?

- No, none of the resistance phenotypes was over-represented among non-typeable isolates.

Except for penicillin and ceftriaxone, the information about susceptibility/resistance to any other beta-lactam(s) seems irrelevant. Also, considering low virulence of NT pneumococci, is information about i.v. formulas of a relevance?

- Thank you for this suggestion. The breakpoints of tested antibiotics were interpreted according to European Committee on Antimicrobial Susceptibility Testing recommendations from 2021 (ref. No 28) which includes other beta-lactam antibiotics and information on i.v. formulas, but we do agree that, considering low virulence of non-typeable pneumococci, this information is of low relevance. In any case, following suggestions of the Reviewer #3, we completely redesigned and reformated theTable with information on antibiotic susceptibility, and focused only on resistance patterns of Streptococcus pneumoniae isolates (Table 4 in the revised version of the manuscript).

Authors reports (line 245) on positive nasopharyngeal or oropharyngeal samples to be serotyped in the study. It seems that S. pneumoniae isolates cultured from samples, and not nucleic acids extracted from the polymicrobial samples were tested in smPCR. Rephrase? 

- Thank you for this suggestion. As we stated in the Material and metod section, we first positive samples were cultured, and then those who were positive serotyped using conventional multiplex PCR assays. 

Describing children living in participants' household as "siblings" is misleading as it implies study subjects were actually brothers or sisters of these children. Also, single-child families cannot be described as families with siblings. 

- We agree with your suggestion and we replaced this with “contact with children aged 0–10 years residing in the household of participants”.

It seems appropriate to describe the study population as “middle-aged and older adults” instead of “adults and elderly adults”. For one, older adults are a subcategory within adults. 

- We agree with your suggestion and we replaced this throughout the whole manuscript.

Line 306: Since serotype 6A is unique for PCV13 and serotypes 11A and 15B for PPS23, should “covered by both” be replaced with “covered by either”?

- Thank you for this. We replaced the word “both” with “either” in that sentence.

Line 129: replace “signs” with “symptoms”? 

- Thank you for this suggestion. We replaced the word “signs” with “symptoms”.

It might be appropriate to consistently use the same term ‘participants’ to describe subjects sampled in the study: replace ‘respondents’in line 212? Table 2 and 3, it does not seem necessary to report on a feature that is universal for all isolates: on vaccination status of carriers inboth tables, and on all isolates listed in Table 2 being cultured from nasopharyngeal samples.

- Thank you for these helpful suggestions. We agree with your opinion and corrected the term “respondent” throughout the manuscript. In addition, based on suggestion from Reviewer #1 the tables 2 and 3 from the previous version were merged into the new one (Table 3 in the present version of the manuscript). Accordingly, reporting on the features that are universal for all isolates was omitted, but information on sample type (NP or OP) was preserved since Spn isolates from both types of samples were presented in the Table 3.

Reviewer #3:

In this manuscript the authors present a study on the pneumococcal carriage from NP and OP in adults ≥ 50 years during COVID-19 in Novi Sad, Serbia. The topic is interesting and of importance as, in contrast to children, there are not many studies in adults. However, there are some issues that need to be dealt with.

Points to be addressed:

1. Authors reported carriage data obtained only by culture-based methods (16 positive samples/1042). Why did they not screen all NP/OP samples for pneumococcal presence using molecular methods as well? It is now agreed that carriage studies should be performed by merging the classical culture-based method with the very sensitive molecular method of qPCR and testing with qPCR the DNA extracted from samples culture-enriched for pneumococci.In addition, it is recommended to include more than one specific target for pneumococcal identification by qPCR, such as lytA, piaB or SP2020, to avoid false positivity due to other viridans streptococci. It is likely that by using molecular methods, carriage rate estimates would be higher. On the other hand, authors performed S. pneumoniae confirmation by lytA and molecular serotyping on isolated streptococci, therefore they can manage this procedure.

-Thank you for this comment. As we have already stated above in response to the Editor and other Reviewers, in designing our study we relied primarily on reccomendations of the World Health Organization working group and tested samples using standard methods provided by these recommendation. In addition, similar methods were also used in a number of other studies with similar design, to which we refer in our manuscript, and some of which had also been published in the PLos One journal (ref. 36. Zanella RC, Brandileone MCC, Almeida SCG, de Lemos APS, Sacchi CT, Gonçalves CR, et al. Nasopharyngeal carriage of Streptococcus pneumoniae, Haemophilus influenzae, and Staphylococcus aureus in a Brazilian elderly cohort. PLoS One. 2019; 14(8):e0221525. doi: 10.1371/journal.pone.0221525. PMID: 31437226; PMCID: PMC6705818; ref. 41.Harimurti K, Saldi SRF, Dewiasty E, Alfarizi T, Dharmayuli M, Khoeri MM, et al. Streptococcus pneumoniae carriage and antibiotic susceptibility among Indonesian pilgrims during the Hajj pilgrimage in 2015. PLoS One. 2021; 16(1):e0246122. doi: 10.1371/journal.pone.0246122. PMID: 33497410; PMCID: PMC7837496; ref. 13. Adetifa IMO, Adamu AL, Karani A, Waithaka M, Odeyemi KA, Okoromah CAN, et al. Nasopharyngeal Pneumococcal Carriage in Nigeria: a two-site, population-based survey. Sci Rep. 2018; 8(1):3509. doi: 10.1038/s41598-018-21837-5. PMID: 29472635; PMCID: PMC5823928) To further support this, we also added reference No 18. in the revised version of the manuscript (Satzke C, Turner P, Virolainen-Julkunen A, Adrian PV, Antonio M, Hare KM, et al; WHO Pneumococcal Carriage Working Group. Standard method for detecting upper respiratory carriage of Streptococcus pneumoniae: updated recommendations from the World Health Organization Pneumococcal Carriage Working Group. Vaccine. 2013;32(1):165-79. doi: 10.1016/j.vaccine.2013.08.062. PMID: 24331112.).

Furthermore, it is worth noting that the prevalence of Spn carriage in our study was not significantly lower than those obtained in other studies with similar methodology. These comparisons were addressed in the Discussion section: “Similar prevalence (3.8%) was found in an Israel study conducted among 1300 screened adults, while prevalence in our study was higher than prevalence of 1.5%, obtained in Slovenian nursing home residents [33], or prevalence of 1.9% obtained from the United States study among retired community residents, or prevalence of 2.2%, among outpatients ≥60 years of age in Brazil, and the prevalence of 2.3% found after testing adults above 60 years of age in Portugal.”

As for the influence of methodology used on the Spn prevalence obtained in our study, we do agree that using molecular methods could have possibly resulted in a higher number of isolates. As the Reviewer has already noticed, we did use PCR to detect lytA to avoid false positivity due to other viridans streptococci. Unfortunately, we could not retest collected samples using PCR in some later timepoint, because the samples have not been preserved. Nevertheless, it should be noted that a low prevalence of Spn carriage among our participants could also be, at least to some extent, due to other reasons that include but are not limited to COVID-19 pandemics and possible effect of non-pharmaceutical interventions and/or social distancing, exclusion of immunocompromised subjects in our study or part of the year when the study was conducted (For more details, please refer to a paragraph on limitations of the study at the end of Discussion section of the manuscript).

Finally, one of the approaches aimed to increase the number of isolates in our study was a decision to include both nasopharyngeal (NP) and oropharyngeal (OP) swabs as recommended by the WHO Pneumococcal Carriage Working Group. As a result, 2 positive samples were from obtained from OP of the participants whose NP were negative for Spn. Similar procedure was used in another study also published in the PLos One journal to which we refer in our manuscript (ref No 3: Almeida ST, Nunes S, Santos Paulo AC, Valadares I, Martins S, Breia F, et al. Low prevalence of pneumococcal carriage and high serotype and genotype diversity among adults over 60 years of age living in Portugal. PLoS One. 2014; 9(3):e90974. doi: 10.1371/journal.pone.0090974. PMID: 24604030; PMCID: PMC3946249.)

2. Authors should describe the Sample Processing (page 5, line 143) in a more detailed way since they report data for both NP and OP but in some cases they refer only to NP samples (see line 149). Since this study is based on NP and OP analysis and 2 positive samples were from OP, authors should use the term nasopharyngeal only if associated with the source of sampling. Accordingly, Authors should refer to “pneumococcal carriage” and not to “nasopharyngeal carriage” throughout the text, tables and in the title of the manuscript.

- Thank you for this remark. In accordance with this suggestions, we have corrected terms and phrasing throughout the entire paper. In addition, the title of the manuscript has also been changed accordingly (the word “nasopharyngeal” has been omitted).

3. In DNA extraction (page 5 line 156) the overnight growth agar plate was that from the enrichment step or was referred to the cultured strain? Please specify.

- DNA was isolated from fresh overnight cultures. In order to clarify that, certain sentences were added in appropriate sections throughout the paper.

4. As per the Serotyping Identification, authors screened for all known serotypes by using Quellung and molecular tests? If no, they should report samples with no identified serotype as Not Determined and not as Non-typeable. Further, in Discussion authors speculate that non-typeable isolates were likely to miss the polysaccharide capsule (lines 264-265), are they sure?

- Thank you for these questions and suggestions. Concerning first question, we should say that in our study Quellung testing was not performed, and that we did serotyping identification using only molecular tests. However, we did screen for all known serotypes that are able to be identified using molecular methods. As for the second question on polysaccharide capsule, all our isolates were tested for cpsA gene in every reaction as recommended by CDC PCR schemes (negative cpsA does not equate to a non-serotypeable isolate). As mentioned in CDC guidelines, positive pneumococcal control band for cpsA is negative in 1-2% of PCR-serotypeable isolates that have been encountered. However, it has been shown that the vast majority of non-typeable isolates by Quellung reaction did not show any amplification for the cpsA internal positive control (ref 25. Jourdain S, Drèze PA, Vandeven J, Verhaegen J, Van Melderen L, Smeesters PR. Sequential multiplex PCR assay for determining capsular serotypes of colonizing S. pneumoniae. BMC Infect Dis. 2011; 11:100. doi: 10.1186/1471-2334-11-100. PMID: 21507244; PMCID: PMC3094224.). Thus, it is plausible to assume that cpsA-negative isolates in our study are at the same time non-typeable as defined by the Quellung reaction. Also, we cannot be absolutely sure that non-typable isolates are missing polysaccharide capsule, but it is likely possibility. Therefore, in our manuscript, we also referred to the findings and conclusions of other studies: for example, authors of the study “Nasopharyngeal carriage of Streptococcus pneumoniae and serotypes indentified among nursing home residents in comparison to the elderly and patients younger than 65 years living in domestic environment. Zdr Varst 2017; 56(3): 172-178. (ref 35)” concluded that “Most of the pneumococci isolated were serotyped, some of them were non-typeable, which means they did not have the polysaccharide capsule.”. In addition, it is known that “Despite its increase in prevalence, little is known about non-typeable “pneumococci. Traditionally, pneumococcus has been identified as alpha-hemolytic colonies that are optochin sensitive and have genes such as lytA and ply. However, these traditional chemical or genetic tests can fail to distinguish pneumococcus from other closely related species in the mitis group, which includes Streptococcus mitis, Streptococcus pseudopneumoniae, and Streptococcus oralis as well as Spn.” (ref. Park IH, Kim KH, Andrade AL, Briles DE, McDaniel LS, Nahm MH. Nontypeable pneumococci can be divided into multiple cps types, including one type expressing the novel gene pspK. mBio. 2012 Apr 24;3(3):e00035-12. doi: 10.1128/mBio.00035-12. PMID: 22532557; PMCID: PMC3340917.). We are aware of this potential problem, but we did not have multilocus sequence typing (MLST) analysis that examines the sequences of 7 different housekeeping gene loci that can be used to reliably distinguish pneumococci from closely related species and they were classified as non-typeable. We believe that in order to reduce the number of non-typeable Spn, future research should include multilocus sequence typing (MLST) analysis whenever possible.

5. In Results which is the importance to describe the characteristics of participants associated with serotyped and not serotyped S. pneumoniae? It is not clear to me.

- Thank you for this question. Our intention was to compare the characteristics of participants with and without determined serotypes of Spn. In the revised version of the manuscript, characteristics of six participants with known serotype of Spn and ten participants who classified as non-typeable Spn were presented in the same table following the suggestion from Reviewer #1. As a result, those characteristics between the two groups were quite similar, as it seems that there were only some difference regarding gender (predominance of female subjects in non-typeable group). Also, participants with determined serotypes of Spn were quite more in contact with children aged 0-10 years than those who harboured non-typeable Spn. It is also interesting that both positive oropharyngeal samples were non-typeable. We are aware that it is a small number of isolates, and that no firm conclusions could be drawn, but we do believe that these preliminary results may serve as basis for future more comprehensive research. 

6. In Results the description of antimicrobial susceptibility pattern is quite confuse, authors report alternatively antibiotic susceptibility rate and antibiotic resistance rate. It should be better to report only the antibiotic resistance rate for each specific class of antibiotics. The same should be corrected in the Abstract.

- Thank you for your suggestion. Although many other authors tend to present both susceptibility and resistence in the same sections/tables (for example: Dagan R, et al. Efficacy of 13-valent pneumococcal conjugate vaccine (PCV13) versus that of 7-valent PCV (PCV7) against nasopharyngeal colonization of antibiotic-nonsusceptible Streptococcus pneumoniae. J Infect Dis. 2015; 211(7):1144-53. doi: 10.1093/infdis/jiu576. Epub 2014 Oct 29. PMID: 25355940; Emgård M, et al. Carriage of penicillin-non-susceptible pneumococci among children in northern Tanzania in the 13-valent pneumococcal vaccine era. Int J Infect Dis. 2019; 81:156-166. doi: 10.1016/j.ijid.2019.01.035. Epub 2019 Jan 24. PMID: 30685588; Hussen S, et al. Pneumococcal nasopharyngeal carriage and antimicrobial susceptibility profile in children under five in southern Ethiopia. F1000Res. 2020; 9:1466. doi: 10.12688/f1000research.27583.3. PMID: 34316364; PMCID: PMC8278251; Manenzhe RI, et al. Nasopharyngeal Carriage of Antimicrobial-Resistant Pneumococci in an Intensively Sampled South African Birth Cohort. Front Microbiol. 2019; 10:610. doi: 10.3389/fmicb.2019.00610. PMID: 30972052; PMCID: PMC6446970.), we do agree that this may be confusing and lead to some misinterpretation of our data. Therefore, we corrected the Table following Reviewer’s suggestion and we kept only data related to resistance pattern of Spn isolates in the new Table 4. Accordingly, the text throughout the whole manuscript, including the Abstract, was also revised.

7. In Results the paragraph related to distribution of Covid-19 and follow-up of participants should be reported only as a discussion point, as a limit of the study.

- Thank you for this suggestion. We do agree that this paragraph could lead to some confusion so we removed data on distribution of COVID-19 cases and a number of sampled participants for Streptococcus pneumoniae during follow-up period from Results section, but, as suggested, referred to this issue with some additional explanation in the paragraph on the limitations of the study at the end of the Discussion section.

The authors would like to thank all the Reviewers and the Editor for the suggestions that helped us improve our paper.

With regards,

Prof Mioljub Ristić

Centre for Disease Control and Prevention, Institute of Public Health of Vojvodina, Novi Sad, Serbia

Futoška 121, Novi Sad 21 000, Serbia

E-mail: mioljub.ristic@mf.uns.ac.rs

---

## [Decision Letter · Decision Letter 1]

3 Aug 2022

PONE-D-22-02347R1Pneumococcal carriage in adults aged 50 years and older in outpatient health care facility during pandemic COVID-19 in Novi Sad, SerbiaPLOS ONE

Dear Dr. Ristić,

Thank you for submitting your manuscript to PLOS ONE. After careful consideration, we feel that it has merit but does not fully meet PLOS ONE’s publication criteria as it currently stands. Therefore, we invite you to submit a revised version of the manuscript that addresses the points raised during the review process.

The manuscript has been assessed by four reviewers. Their comments are available below. Two reviewers have raised a number of concerns about the manuscript and also about the quality of the english.

Please carefully revise the manuscript to address all the points raised by the two reviewers.

We look forward to receiving your revised manuscript.

Kind regards,

Jose Melo-Cristino, M.D., Ph.D.

Academic Editor

PLOS ONE

Reviewers' comments:

Reviewer's Responses to Questions

**Comments to the Author**

1. If the authors have adequately addressed your comments raised in a previous round of review and you feel that this manuscript is now acceptable for publication, you may indicate that here to bypass the “Comments to the Author” section, enter your conflict of interest statement in the “Confidential to Editor” section, and submit your "Accept" recommendation.

Reviewer #1: All comments have been addressed

Reviewer #2: All comments have been addressed

Reviewer #3: (No Response)

Reviewer #4: (No Response)

2. Is the manuscript technically sound, and do the data support the conclusions?

Reviewer #1: Partly

Reviewer #2: Yes

Reviewer #3: Yes

Reviewer #4: Yes

3. Has the statistical analysis been performed appropriately and rigorously? 

Reviewer #1: Yes

Reviewer #2: Yes

Reviewer #3: Yes

Reviewer #4: Yes

4. Have the authors made all data underlying the findings in their manuscript fully available?

Reviewer #1: Yes

Reviewer #2: Yes

Reviewer #3: Yes

Reviewer #4: Yes

5. Is the manuscript presented in an intelligible fashion and written in standard English?

Reviewer #1: Yes

Reviewer #2: Yes

Reviewer #3: Yes

Reviewer #4: Yes

6. Review Comments to the Author

Reviewer #1: The manuscript by Ristic and colleagues has been significantly improved, regarding both clarity and scientific rigor. The statistical analysis has been improved and the presentation of the results is clearer. Although the main limitation remains, the effort made by the authors to explain the rational for the study and to acknowledge its limitations is very well reflected in this version of the manuscript.

Reviewer #2: (No Response)

Reviewer #3: This is a revised version of the manuscript “Pneumococcal carriage in adults aged 50 years and older in outpatient health care facility during pandemic COVID-19 in Novi Sad, Serbia.

In general, the overall revision, especially for the editing of scientific language and corrections/implementations according to some requests, improved the quality of the manuscript. However, the most critical issue concerning the lack of molecular methods was not solved due to non-availability of original specimens. Authors stated that design of the study was based on the WHO recommendations (ref 18) that were published in 2014. Indeed, during the latest years a number of other studies gave indications on best procedure for pneumococcal carriage studies, especially in adults, such as “Arguedas A, Trzciński K, O'Brien KL, Ferreira DM, Wyllie AL, Weinberger D, Danon L, Pelton SI, Azzari C, Hammitt LL, Sá-Leão R, Brandileone MC, Saha S, Suaya J, Isturiz R, Jodar L, Gessner BD. Upper respiratory tract colonization with Streptococcus pneumoniae in adults. Expert Rev Vaccines. 2020 Apr;19(4):353-366. doi: 10.1080/14760584.2020.1750378. Epub 2020 Apr 17. PMID: 32237926”.

Reviewer #4: The authors have addressed most of the comments raised by the reviewers, offered mostly satisfactory responses and provided necessary information in the text, including acknowledgement of the study limitations. A few issues remain that would deserve the attention of the authors.

1) Lines 110-111. Please delete this sentence since it remains unclear, despite the authors’ previous response, how the information presented could serve to support the use or not of existing pneumococcal vaccines in adults.

2) Line 122. In English it should be “0.4%”.

3) Lines 157-158, “The primary outcomes were colonization by Spn (culture and multiplex PCR) and Staphylococcus aureus (culture)”, please change to “The primary outcomes were colonization by Spn (culture) and Staphylococcus aureus (culture)”. The current sentence can be misleading, since it can be interpreted to mean that colonization was sough by molecular methods, which is not the case and constitutes a major limitation of the study.

4) Line 174. Typographical error, should be “gene”.

5) Line 175. lytA should be in italic.

6) Lines 179-181. Can the authors specifically state how many serotypes could have been identified by their PCR schema? This is important to interpret their “non-typeable” isolates.

7) Table 1. The information under the heading “Carriage of Streptococcus pneumoniae” is redundant with that under “Swab positive for Streptococcus pneumoniae” and should be eliminated.

8) Table 1. The heading “Received vaccines (PCV13 or PPSV23 )” should be changed to “Received PCV13” since no participant received PPSV23.

9) Lines 232-233. Please delete “with female predomination (Female vs. Male: 7 vs. 3)” since, as far as I understood, this is not a statistically supported difference.

10) Table 3. Typographical error in the heading, should read: “Non-typeable Streptococcus pneumoniae isolates”

11) Line 255. Replace “actual” for “the current”

12) Line 261. Please delete “(boosted by the presence of Spn in their children)”, since this is a strange sentence construction that may be misunderstood for the fact that children themselves boost antibody production.

13) Line 276, Please replace “probably” for “possibly”.

14) Lines 285-287, sentence starting with “The serotype distribution among our participants was not affected by the reception of vaccine (…)”. Please delete the sentence since the fact that only two participants were vaccinated precludes any conclusion regarding the effect of vaccination on carriage.

7. PLOS authors have the option to publish the peer review history of their article (what does this mean?). If published, this will include your full peer review and any attached files.

Reviewer #1: No

Reviewer #2: No

Reviewer #3: No

Reviewer #4: No

---

## [Author Response · Author response to Decision Letter 1]

10 Aug 2022

PONE-D-22-02347R1

Pneumococcal carriage in adults aged 50 years and older in outpatient health care facility during pandemic COVID-19 in Novi Sad, Serbia

PLOS ONE

Dear Dr. Ristić,

Thank you for submitting your manuscript to PLOS ONE. After careful consideration, we feel that it has merit but does not fully meet PLOS ONE’s publication criteria as it currently stands. Therefore, we invite you to submit a revised version of the manuscript that addresses the points raised during the review process.

The manuscript has been assessed by four reviewers. Their comments are available below. Two reviewers have raised a number of concerns about the manuscript and also about the quality of the english.

We look forward to receiving your revised manuscript.

Kind regards,

Jose Melo-Cristino, M.D., Ph.D.

Academic Editor

PLOS ONE

Dear Editor,

First of all, we would like to thank you for considering our paper for publication in your Journal. We are grateful for all the suggestions and remarks, which we find well-intended and helpful. The authors have considered all the comments and we did our best to correct the manuscript in the most appropriate and clear way. All the changes that have been made are highlighted in the manner proposed by the Journal. 

Reviewers' comments:

Reviewer's Responses to Questions

Comments to the Author

1. If the authors have adequately addressed your comments raised in a previous round of review and you feel that this manuscript is now acceptable for publication, you may indicate that here to bypass the “Comments to the Author” section, enter your conflict of interest statement in the “Confidential to Editor” section, and submit your "Accept" recommendation.

Reviewer #1: All comments have been addressed

Reviewer #2: All comments have been addressed

Reviewer #3: (No Response)

Reviewer #4: (No Response)

2. Is the manuscript technically sound, and do the data support the conclusions?

Reviewer #1: Partly

Reviewer #2: Yes

Reviewer #3: Yes

Reviewer #4: Yes

3. Has the statistical analysis been performed appropriately and rigorously?

Reviewer #1: Yes

Reviewer #2: Yes

Reviewer #3: Yes

Reviewer #4: Yes

4. Have the authors made all data underlying the findings in their manuscript fully available?

Reviewer #1: Yes

Reviewer #2: Yes

Reviewer #3: Yes

Reviewer #4: Yes

5. Is the manuscript presented in an intelligible fashion and written in standard English?

Reviewer #1: Yes

Reviewer #2: Yes

Reviewer #3: Yes

Reviewer #4: Yes

6. Review Comments to the Author

Response to Reviewers

IMPORTANT!

Comments to the Author are shown in regular fonts, while the comments to the Reviewers and the academic Editor are in the italic fonts.

Reviewer #1: The manuscript by Ristic and colleagues has been significantly improved, regarding both clarity and scientific rigor. The statistical analysis has been improved and the presentation of the results is clearer. Although the main limitation remains, the effort made by the authors to explain the rational for the study and to acknowledge its limitations is very well reflected in this version of the manuscript.

Thank you very much for these comments. Though we strived to overcome all the limitations of our study, some of them still exist, but we sincerely hope that they have all been addressed appropriately in the present version of the manuscript.

Reviewer #2: (No Response)

Reviewer #3: This is a revised version of the manuscript “Pneumococcal carriage in adults aged 50 years and older in outpatient health care facility during pandemic COVID-19 in Novi Sad, Serbia.

In general, the overall revision, especially for the editing of scientific language and corrections/implementations according to some requests, improved the quality of the manuscript. However, the most critical issue concerning the lack of molecular methods was not solved due to non-availability of original specimens. Authors stated that design of the study was based on the WHO recommendations (ref 18) that were published in 2014. Indeed, during the latest years a number of other studies gave indications on best procedure for pneumococcal carriage studies, especially in adults, such as “Arguedas A, Trzciński K, O'Brien KL, Ferreira DM, Wyllie AL, Weinberger D, Danon L, Pelton SI, Azzari C, Hammitt LL, Sá-Leão R, Brandileone MC, Saha S, Suaya J, Isturiz R, Jodar L, Gessner BD. Upper respiratory tract colonization with Streptococcus pneumoniae in adults. Expert Rev Vaccines. 2020 Apr;19(4):353-366. doi: 10.1080/14760584.2020.1750378. Epub 2020 Apr 17. PMID: 32237926”.

Thank you for these observations. Among many others, we also carrefully read the aforementioned reference. In this publication authors highlighted that the prevalence of Spn in adults is low. They suggested that the ‘pediatric approach’ in Spn colonization determination may be insufficient in adults and that the pneumococcal detection in this population may be improved by longitudinal studies that include samples from additional respiratory sites combined with more extensive laboratory testing that include molecular techniques in addition to sensitive culture-based methods. In order to improve the surveillance of Spn in adults, authors of this study also proposed that in pneumococcal carriage studies, samples should be collect not only from nasopharynx, but also from alternative respiratory sites such as the oropharynx, saliva, or nasal wash and that the culture enrichment for pneumococcus should be used.

As already noted, we were not able to perform the qPCR in our study due to non-availability of original specimens. However, in order to increase the sensitivity of our study, we collected oropharyngeal swabs in parallel to nasopharingeal swabs from each participant and used culture enrcihment for Spn in all our samples, regardless of their origin. As a result, we identified two participant that harboured Spn only in their oropharynx. We would also like to point out that, despite the absence of the molecular method for further detection of our specimens, the prevalence of carriage of Spn in adults in our study was comparable with or even higher than the prevalences found in several other studies that had been performed in other countries from different geographical areas (references 3, 7, 9, 35 and 36 in our manuscript). This issue was adressed in the Discussion section of the manuscript. Thus, although we are fully aware of the limitations of our study, we do believe that our study added some important additional information that can be of interest for the wider scientific community.

Reviewer #4: The authors have addressed most of the comments raised by the reviewers, offered mostly satisfactory responses and provided necessary information in the text, including acknowledgement of the study limitations. A few issues remain that would deserve the attention of the authors.

1) Lines 110-111. Please delete this sentence since it remains unclear, despite the authors’ previous response, how the information presented could serve to support the use or not of existing pneumococcal vaccines in adults.

Thank you for this suggestion. We removed this sentence in the present version of the manuscript.

2) Line 122. In English it should be “0.4%”.

This was corrected.

3) Lines 157-158, “The primary outcomes were colonization by Spn (culture and multiplex PCR) and Staphylococcus aureus (culture)”, please change to “The primary outcomes were colonization by Spn (culture) and Staphylococcus aureus (culture)”. The current sentence can be misleading, since it can be interpreted to mean that colonization was sough by molecular methods, which is not the case and constitutes a major limitation of the study.

Thank you for this helpful suggestion. We changed the sentence as proposed by the Reviewer.

4) Line 174. Typographical error, should be “gene”.

We replaced incorrectly spelled word „gen“ with „gene“.

5) Line 175. lytA should be in italic.

This was corrected as suggested.

6) Lines 179-181. Can the authors specifically state how many serotypes could have been identified by their PCR schema? This is important to interpret their “non-typeable” isolates.

According to the CDC recommended schemes and primers for pneumococcal serotype deduction, it is possible to identify 41 different serotypes of Spn. We added this fact appropriately in the text of the present version of the manuscript.

7) Table 1. The information under the heading “Carriage of Streptococcus pneumoniae” is redundant with that under “Swab positive for Streptococcus pneumoniae” and should be eliminated.

As suggested, the Table 1 was corrected and the redundant data removed.

8) Table 1. The heading “Received vaccines (PCV13 or PPSV23 )” should be changed to “Received PCV13” since no participant received PPSV23.

Thank you for this suggestion. In the Table 1, we corrected the heading with „Received PCV13“. 

9) Lines 232-233. Please delete “with female predomination (Female vs. Male: 7 vs. 3)” since, as far as I understood, this is not a statistically supported difference.

We agree with your suggestion. There was not statistically significant difference regarding gender, and therefore we removed this part from the sentence.

10) Table 3. Typographical error in the heading, should read: “Non-typeable Streptococcus pneumoniae isolates”

This was corrected as suggested.

11) Line 255. Replace “actual” for “the current”

The replacement was done.

12) Line 261. Please delete “(boosted by the presence of Spn in their children)”, since this is a strange sentence construction that may be misunderstood for the fact that children themselves boost antibody production.

We agree with your suggestion and we removed this part of the sentence.

13) Line 276, Please replace “probably” for “possibly”.

It was replaced.

14) Lines 285-287, sentence starting with “The serotype distribution among our participants was not affected by the reception of vaccine (…)”. Please delete the sentence since the fact that only two participants were vaccinated precludes any conclusion regarding the effect of vaccination on carriage.

We absolutely agree with this, so we removed this part of the sentence and accordingly rephrased the whole sentence in the present version of the manuscript.

Sincerely, 

Mioljub Ristić, MD, PhD

Centre for Disease Control and Prevention, Institute of Public Health of Vojvodina, Novi Sad, Serbia

Futoška 121, Novi Sad 21 000, Serbia

E-mail: mioljub.ristic@mf.uns.ac.rs

---

## [Editor Report · Decision Letter 2]

22 Aug 2022

PONE-D-22-02347R2Pneumococcal carriage in adults aged 50 years and older in outpatient health care facility during pandemic COVID-19 in Novi Sad, SerbiaPLOS ONE

Dear Dr. Ristić,

Thank you for submitting your manuscript to PLOS ONE. After careful consideration, we feel that it has merit but does not fully meet PLOS ONE’s publication criteria as it currently stands. Therefore, we invite you to submit a revised version of the manuscript that addresses the points raised during the review process.

Thank you for providing the last review of the manuscript. One important issue remains to clarify, concerning antimicrobial susceptibility. In Materials and Methods you mention that disc diffusion technique was used according to EUCAST with no further details (but including a correct reference). In Results you report one isolate resistant to amoxicillin (per os) and amoxicillin-clavulanic acid (per os) but susceptible to penicillin and to ampicillin. This result is not in accordance to the rules of EUCAST and should be corrected or further explained how the result was obtained.A second issue (less important) is: in Table 1, values of % should have only one decimal, not two.

We look forward to receiving your revised manuscript.

Kind regards,

Jose Melo-Cristino, M.D., Ph.D.

Academic Editor

PLOS ONE
---

## [Author Response · Author response to Decision Letter 2]

31 Aug 2022

PONE-D-22-02347R2

Pneumococcal carriage in adults aged 50 years and older in outpatient health care facility during pandemic COVID-19 in Novi Sad, Serbia

PLOS ONE

Dear Dr. Ristić,

Thank you for submitting your manuscript to PLOS ONE. After careful consideration, we feel that it has merit but does not fully meet PLOS ONE’s publication criteria as it currently stands. Therefore, we invite you to submit a revised version of the manuscript that addresses the points raised during the review process.

Thank you for providing the last review of the manuscript. One important issue remains to clarify, concerning antimicrobial susceptibility. In Materials and Methods you mention that disc diffusion technique was used according to EUCAST with no further details (but including a correct reference). In Results you report one isolate resistant to amoxicillin (per os) and amoxicillin-clavulanic acid (per os) but susceptible to penicillin and to ampicillin. This result is not in accordance to the rules of EUCAST and should be corrected or further explained how the result was obtained.

A second issue (less important) is: in Table 1, values of % should have only one decimal, not two.

We look forward to receiving your revised manuscript.

Kind regards,

Jose Melo-Cristino, M.D., Ph.D.

Academic Editor

PLOS ONE

Dear Editor,

We would like to thank you, once again, for considering our paper for publication in your Journal. We would also like to thank you and the Reviewers for the well-intended suggestions and remarks, which helped us to improve our manuscript. We have considered them and corrected, as suggested. All the changes that have been made are highlighted in the manner proposed by the Journal. Comments to the Authors are shown in regular fonts, while the responses to the academic Editor (and the Reviewers) are in the italic fonts, as follows:

Editor:

One important issue remains to clarify, concerning antimicrobial susceptibility. In Materials and Methods you mention that disc diffusion technique was used according to EUCAST with no further details (but including a correct reference). In Results you report one isolate resistant to amoxicillin (per os) and amoxicillin-clavulanic acid (per os) but susceptible to penicillin and to ampicillin. This result is not in accordance to the rules of EUCAST and should be corrected or further explained how the result was obtained

Thank you for this important observation, which relates to our unintentional mistake that could be misleading. Indeed, we have determined antimicrobial susceptibility of our isolates using disc diffusion technique according to EUCAST, and found one non-typeable isolate that was resistant to amoxicillin and amoxicillin-clavulanic acid, but it was also resistant to penicillin and ampicillin, as well. However, the data on penicillin and ampicillin resistance have been inadvertently omitted from Table 4 in the original version of the manuscript, and, consequently, the susceptibility pattern of this isolate was not properly reported. Therefore, in the present version of the manuscript, we have changed Table 4 to include correct results of susceptibility of our isolates. Accordingly, the statements in the Abstract, Result section on Antimicrobial resistance patterns of Streptococcus pneumoniae and in Discussion section referring to resistance of isolates to Beta-lactam antibiotics were also changed.

Editor:

A second issue (less important) is: in Table 1, values of % should have only one decimal, not two.

As suggested, Table 1 has been corrected in the revised version of the manuscript and all values of percentages now contain only one decimal.

Sincerely, 

Mioljub Ristić, MD, PhD

Centre for Disease Control and Prevention, Institute of Public Health of Vojvodina, Novi Sad, Serbia

Futoška 121, Novi Sad 21 000, Serbia

E-mail: mioljub.ristic@mf.uns.ac.rs

---

## [Editor Report · Decision Letter 3]

2 Sep 2022

Pneumococcal carriage in adults aged 50 years and older in outpatient health care facility during pandemic COVID-19 in Novi Sad, Serbia

PONE-D-22-02347R3

Dear Dr. Ristić,

We’re pleased to inform you that your manuscript has been judged scientifically suitable for publication and will be formally accepted for publication once it meets all outstanding technical requirements.

Kind regards,

Jose Melo-Cristino, M.D., Ph.D.

Academic Editor

PLOS ONE
---

## [Editor Report · Acceptance letter]

2 Oct 2022

PONE-D-22-02347R3 

Pneumococcal carriage in adults aged 50 years and older in outpatient health care facility during pandemic COVID-19 in Novi Sad, Serbia 

Dear Dr. Ristić:

I'm pleased to inform you that your manuscript has been deemed suitable for publication in PLOS ONE. Congratulations! Your manuscript is now with our production department. 

Kind regards, 

on behalf of

Prof. Jose Melo-Cristino 

Academic Editor

PLOS ONE